# Submandibular Push Exercise Using Visual Feedback from a Pressure Sensor in Patients with Swallowing Difficulties: A Pilot Study

**DOI:** 10.3390/healthcare9040407

**Published:** 2021-04-01

**Authors:** Jong-Moon Hwang, Hyunwoo Jung, Chul-Hyun Kim, Yang-Soo Lee, Myunghwan Lee, Soo Yeon Hwang, Ae-Ryoung Kim, Donghwi Park

**Affiliations:** 1Department of Rehabilitation Medicine, Kyungpook National University Hospital, Daegu 41944, Korea; hti82@hanmail.net (J.-M.H.); hwjung_87@naver.com (H.J.); chgim@knu.ac.kr (C.-H.K.); leeyangsoo@knu.ac.kr (Y.-S.L.); 2Department of Rehabilitation Medicine, School of Medicine, Kyungpook National University, Daegu 41944, Korea; 3Medical Device Development Center, Daegu-Gyeongbuk Medical Innovation Foundation, Daegu 41061, Korea; lmh0107@dgmif.re.kr (M.L.); mungnim@dgmif.re.kr (S.Y.H.); 4Department of Physical Medicine and Rehabilitation, Ulsan University Hospital, University of Ulsan College of Medicine, Ulsan 44033, Korea

**Keywords:** submandibular push exercise, suprahyoid, infrahyoid, CTAR, Shaker exercise

## Abstract

*Objectives:* We aimed to determine the usefulness and effectiveness of a submandibular push exercise with visual feedback from a pressure sensor in patients with dysphagia through continuous exercise sessions. *Methods:* Twelve patients with dysphagia of various etiologies were included. A total of five exercise sessions (every 3 or 4 days) over three weeks were conducted. During the submandibular push exercise, patients were instructed to maintain a maximum force for 3 s, repeated for 1 min to measure the number of exercises, the maximum pressure, and the area of the pressure-time graph. We statistically compared the values of each exercise trial. *Results:* Among the 12 patients, eight completed the exercise sessions. As the number of exercise trials increased, the maximum pressure and the area in the pressure-time graph showed a significant increase compared to the previous attempt (*p* < 0.05). The maximum pressure and the area of the pressure-time graph improved from the first to the fourth session (*p* < 0.05). The values were maintained after the fourth session, and there was no significant difference between the fourth and the fifth exercise (*p* > 0.05). There was no significant difference between successful and non-successful groups, except for the Modified Barthel Index (*p* < 0.05). *Conclusion:* Through repetitive exercise training, the submandibular push exercise using visual feedback from a pressure sensor can be applied as an exercise method to strengthen swallowing related muscles, such as the suprahyoid and infrahyoid muscles. However, additional studies including more patients and a long-term study period are warranted to evaluate the effects of the exercise for improvement of dysphagia.

## 1. Introduction

Swallowing is a complex motor process that involves a variety of muscles of the mouth, as well as the tongue, larynx, pharynx, and esophagus [1,2,3]. Malfunctions of the central nervous system, deconditioning due to various reasons, and degenerative changes may affect movement and coordination of the muscles involved in the swallowing function, leading to dysphagia in the elderly [4,5,6,7,8]. Among the variety of muscles that are involved in swallowing, the importance of the suprahyoid and infrahyoid muscles has been extensively reported in previous research [6]. The suprahyoid muscle is responsible for moving the hyoid bone in an anterosuperior direction [4,9,10,11,12]. Infrahyoid muscles, such as the thyrohyoid, sternohyoid, omohyoid, and sternothyroid muscles, move the larynx in an anterosuperior direction and depress the hyolaryngeal complex [9,10,11]. For these reasons, in the clinical setting, various exercises which can strengthen the suprahyoid and infrahyoid muscles, such as the Shaker and chin tuck against resistance (CTAR) exercises, have been applied to improve swallowing function [13,14,15]. The Shaker exercise is reported to strengthen the suprahyoid and infrahyoid muscles for opening the upper esophageal sphincter (UES) [15,16]. Similarly, the CTAR exercise may also strengthen these muscles by compression of an rubber ball or plastic bar between the chin and the sternum [14,15]. However, the main motion of both exercises is neck flexion, which is not the main function of the suprahyoid and infrahyoid muscles, but rather of the sternocleidomastoid (SCM) muscle. In a previous study [17], a submandibular push exercise was proposed for selective strengthening of the suprahyoid and infrahyoid muscles because it does not include neck flexion, which is the main motion of SCM. Although a submandibular push exercise is reported to efficiently induce selective contraction of the suprahyoid and infrahyoid muscles without less involvement of SCM, it is difficult for a patient to understand how to perform the contraction. To overcome this, visual feedback from a pressure sensor was used to show the patient whether the exercise was being performed effectively; however, despite the usefulness of this feedback, it was only evaluated during one session in the previous study [17].

Therefore, in this study, we aimed to investigate the usefulness and effectiveness of submandibular push exercises with visual feedback from a pressure sensor in patients with dysphagia through continuous exercise sessions for 3 weeks.

## 2. Materials and Methods

### 2.1. Participants

This study was approved by the Kyungpook National University College of Medicine (No. KNUMC_2019_04_015) and informed consent was obtained from the participants. In this prospective study conducted between May 2019 and October 2020, 12 adult participants with dysphagia and underlying diseases such as stroke, spinal cord injury, and deconditioning were initially recruited (Table 1). The patients had various dysphagia etiologies with at least one symptom, such as food sticking in the throat, coughing when eating, globus sensation, drooling, a weak or wet voice, and difficulty in chewing [18,19]. These patients were finally diagnosed with dysphagia through Videofluoroscopic Swallowing Study (VFSS). We evaluated the VFSS with Videofluoroscopic Dysphagia Scale (VDS) and Penetration–Aspiration Scale (PAS) by 2 physiatrists (Jong-moon Hwang, Hyunwoo Jung). The VDS is a evaluation scale that access comprehensive swallowing function in stroke patients based on VFSS. The VDS include 14 items that represent the oral phase (lip closure, bolus formation, mastication, apraxia, tongue to palate contact, premature bolus loss and oral transit time) and pharyngeal phase (triggering of pharyngeal swallowing, vallecular residue, pyriform sinus residue, laryngeal elevation, coating of the pharyngeal wall, pharyngeal transit time and aspiration). The VDS is rated from 0 to 100, and the higher the score, the higher the severity of dysphagia [20]. The PAS is a standard scale that represents laryngeal penetration and aspiration. Penetration is defined as passage of test materials (yogurt, water) into the larynx, which do not go through below the vocal folds, while aspiration means the passage of test materials entering the airway below the vocal folds. The scale is separated into eight levels based on the extents of test materials passage into the airway, with higher levels representing the higher dysphagia severity [21]. All patients had stable vital signs and were able to obey our instructions to perform the submandibular push exercise. Patients with severe cognitive dysfunction (≤9 points on the Mini-Mental Status Examination (MMSE)), serious psychiatric disorders, other problems that may have limited the use of the devices during the exercise, or those aged less than 20 years, were excluded [18].

### 2.2. The Method of Submandibular Push Exercise

It is difficult for the patient to precisely do the submandibular push exercise. So, we induced a submandibular push exercise through three tongue movements. First, with the patient’s tongue pulled back, the bottom of the tongue was strongly pressed against the bottom of the mouth. Second, the patient’s tongue was turned over and the roof of the mouth was pressed strongly against the underside of the tongue. Third, the patient’s tongue was turned over and the tip of the tongue was directed to the back of the neck as far as possible (Figure 1). Among these three methods, we trained one method that the patient did the best.

### 2.3. Submandibular Push Exercise Using Visual Feedback from a Pressure Sensor

Patients were shown how to perform the exercise [17] by a physiatrist. Patients applied a headgear with pressure sensor placed under the suprahyoid muscles. Patients were asked to move the tongue and maintain the highest submandibular pressure with the visual feedback through the pressure sensor graph displayed on the computer monitor in real time. After the demonstration, the patient had a 30 min training session under the supervision of the investigators. Once the patient was familiar with the exercise, five sessions (every 3 or 4 days) were conducted for 3 weeks.

Patients could practice for 10 min at the start of each session. Then, after a sufficient rest of 10 min or more, a 1 min exercise session was started. During the submandibular push exercise, patients were instructed to maintain maximum force for 3 s, and then patients relaxed the muscle. 3 s after the relaxing the muscle, we gave the que for maintain maximum force again. Repeating this cycle for 1 min to measure the total exercise performance (total area of the pressure graph during 1 min) and the maximum pressure (Figure 2).

We included the patients in the success group who precisely contracted the suprahyoid and infrahyoid muscles and delivered the force to the pressure sensor by following our instructions. Additionally, we sorted the patients in the failure group who could not perform the exercise and deliver the force to the pressure sensor exactly as directed by our instructions due to physical and cognitive impairment caused by each etiologies.

Then, we analyzed the difference in age, days of disease duration, and initial behavior and functional parameters (Berg Balance Scale (BBS), Korean version of Mini-Mental State Examination (MMSE-K), Modified Barthel Index (MBI), Penetration–Aspiration Scale (PAS), and Videofluoroscopic Dysphagia Scale (VDS)) between the two groups. Berg balance scale consists of a 14-items that quantitatively evaluates sitting, standing and gait balance and risk for falls through their performance [22]. Modified Barthel Index measures the individual’s activities of daily living function in 10 items (personal hygiene, bathing self, feeding, toilet, stair climbing, dressing, bowel control, bladder control, ambulation, chair/bed transfer) [23].

### 2.4. Statistical Analysis

IBM SPSS version 21 (SPSS, Inc., Chicago, IL, USA) was used for the statistical analysis. To determine whether significant differences existed among the five sessions, the values of the number of exercises, maximum pressure, and area of pressure-time graph for the participants were compared using repeated Wilcoxon signed rank test. *p* < 0.05 was considered statistically significant.

## 3. Results

### 3.1. Participant Characteristics

A total of 12 subjects (11 males and one female), all of whom underwent inpatient treatment at our hospital’s rehabilitation department, participated in the study. Seven patients suffered ischemic or hemorrhagic stroke, three had spinal cord injuries, and two presented with a deconditioned state due to cardiac arrest and multiple trauma (Table 1).

### 3.2. Comparison of Success and Non-Success Group

Of the 12 subjects, eight succeeded in the submandibular push exercise and four failed. We analyzed the difference in age, days of disease duration, and initial behavior and functional parameters [Berg Balance Scale (BBS), Korean version of Mini-Mental State Examination (MMSE-K), Modified Barthel Index (MBI), Penetration–Aspiration Scale (PAS), and Videofluoroscopic Dysphagia Scale (VDS)] between the two groups (Table 1). There was a significant difference in the initial BBS (*p* < 0.05). There was no statistically significant difference in initial MBI, but there was a significant difference in mean MBI values between the success group and the failure group.

### 3.3. Comparison of Each Exercise Trial

Patients performed five exercise sessions using the submandibular push exercise with visual feedback. To access the performance of the submandibular push exercise, we investigated the number of exercises, the maximum pressure, and the area of the pressure-time graph and statistically compared the values of each trial (Table 2). As the number of exercise trials increased, the maximum pressure and the area of the pressure-time graph showed a significant increase compared to the previous attempt (*p* < 0.05), but the number of exercises did not change significantly even after the exercise trial (Figure 3, *p* > 0.05). The maximum pressure and the area of the pressure-time graph improved from the first to the fourth session (*p* < 0.05); however, these values were consistent thereafter, and there was no significant difference between the fourth and the fifth session (Figure 4 and Figure 5, *p* > 0.05).

## 4. Discussion

Disease severity affects the understanding and performance of the submandibular push exercise and the performance of the exercise improved from session one to session four. During the exercise, four out of 12 patients were unable to proceed. Between the successful and non-successful groups, only MBI was a significant factor. This clearly shows that functional level is important to conduct the submandibular push exercise. Patients with a decreased functional level often tend to have medical problems and a decreased condition [24,25,26,27]. For example, medical problems such as pneumonia, urinary tract infections, and pressure sores that occur in patients with low functional levels, worsen the patient’s condition and render participation in the exercise difficult. Although there were few participants in the present study, cognition did not show a significant difference between the success and non-success groups, meaning that mild cognitive impairment may be overcome over several training sessions.

Regarding the effect of the submandibular push exercise with visual feedback, the more effective exercises were performed from the first to the fourth training sessions, a greater number of exercises were performed, and greater maximum pressure was exerted. Although the number of patients was few, most performed well after the fourth session; this may mean that at least four exercise sessions are required for this exercise to be effective. This means that by further continuation of the submandibular push exercise with visual feedback, the function of the suprahyoid and infrahyoid can be improved. With consideration of the results of the previous study [17], the amount of pressure in the submandibular area is proportional to the activity of the suprahyoid muscle contraction. Therefore, the submandibular push exercise, performed in conjunction with the CTAR and Shaker exercises [14,15,16], could provide patients with more options to strengthen the suprahyoid and infrahyoid muscles.

There were a few limitations in our study. First, the sample size was small and our study did not run for a long period. Second, we did not demonstrate an improvement in swallowing function through pre- and post-swallowing tests, such as VFSS; however, the maximal value of pressure, which is known to indirectly reflect the activity of the suprahyoid muscle contraction, and the total pressure value were certainly improved as sessions progressed. This suggests that the submandibular push exercise may improve swallowing difficulty by strengthening the suprahyoid and infrahyoid muscles. In addition, even patients who did not initially perform the submandibular push exercise properly showed improved performance through continuous visual feedback. Therefore, we believe that our study provides sufficient worth as a preliminary study. Additional studies inclusive of more patients and longer study periods are warranted to investigate the effect of the submandibular push exercise for improvement of dysphagia.

## 5. Conclusions

Through repetitive exercise training, the submandibular push exercise using visual feedback from a pressure sensor can be applied as an exercise method to strengthen swallowing related muscles, such as the suprahyoid and infrahyoid muscles. However, additional studies including more patients and a long-term study period are warranted to evaluate the effects for improvement of dysphagia.

## Figures and Tables

**Figure 1 healthcare-09-00407-f001:**
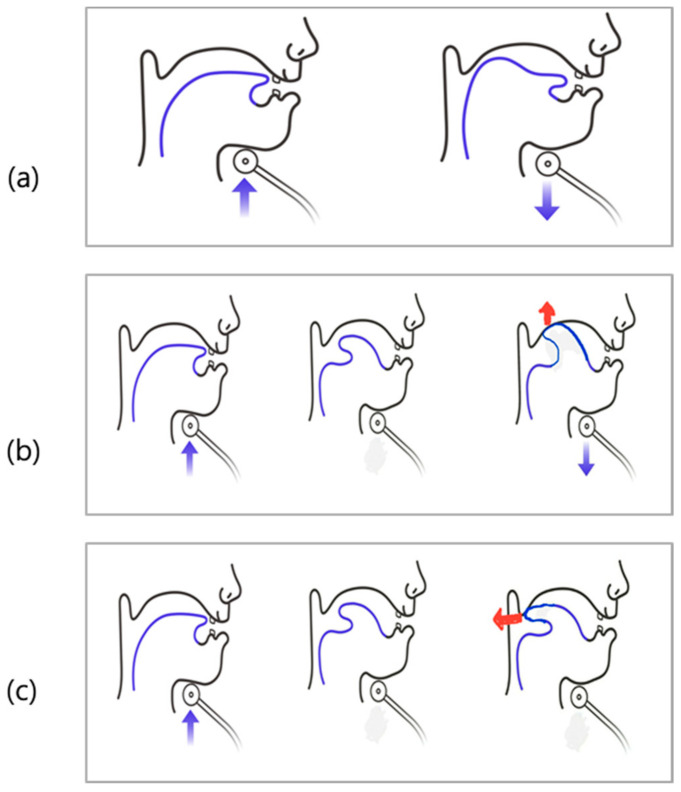
Three tongue motions for submandibular push exercise. (**a**) With the tongue pulled back, the bottom of the tongue was strongly pressed against the bottom of the mouth. (**b**) Tongue was turned over and the roof of the mouth was pressed strongly against the underside of the tongue. (**c**) Tongue was turned over and the tip of the tongue was directed to the back of the neck as far as possible.

**Figure 2 healthcare-09-00407-f002:**
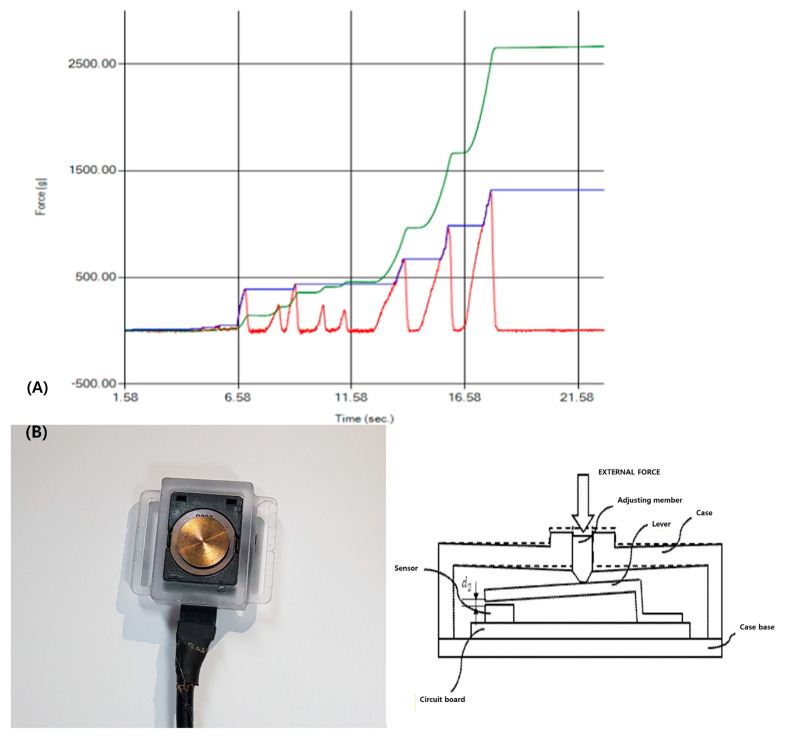
(**A**) A graph explaining the pressure force over time during submandibular push exercise. Redline, the force applied to the sensor over time; Blueline, the maximum force during the measurement time; Greenline, the integral value of force-time graph. (**B**) Load cell pressure sensor and monitoring of pressure sensor during the submandibular push exercise, which used visual feedback [14].

**Figure 3 healthcare-09-00407-f003:**
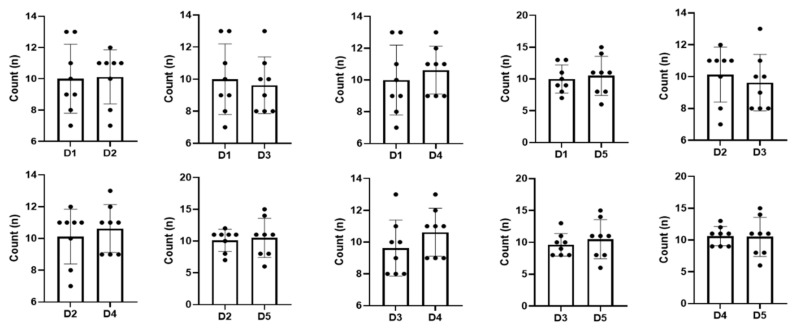
Comparison of the number of exercises for each trial. Count—the number of exercises; D1, 2, 3, 4, 5—dysphagia therapy (submandibular push exercise) trial times. The dots in the graph represent values(count) for each patient.

**Figure 4 healthcare-09-00407-f004:**
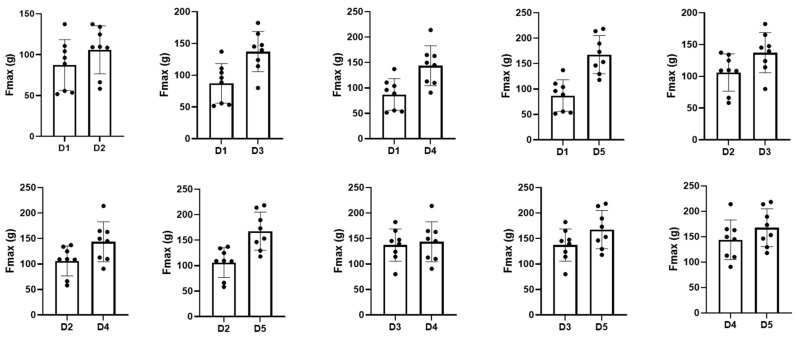
Comparison of the maximum pressure for each trial. Fmax—maximum pressure; D1, 2, 3, 4, 5—dysphagia therapy (submandibular push exercise) trial times. The dots in the graph represent values(Fmax) for each patient.

**Figure 5 healthcare-09-00407-f005:**
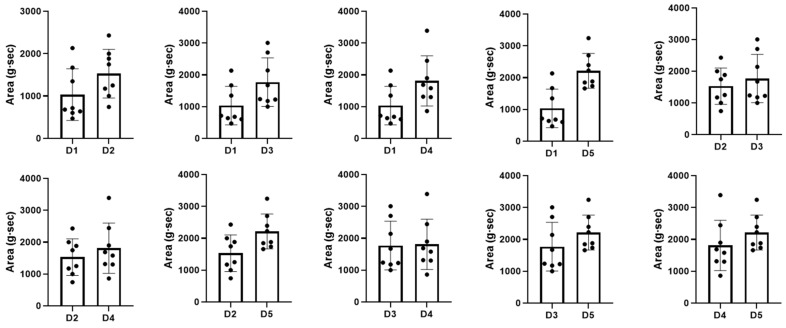
Comparison of the area of pressure-time graph for each trial. Area—the area of pressure-time graph; D1, 2, 3, 4, 5—dysphagia therapy (submandibular push exercise) trial times. The dots in the graph represent values (area of pressure-time graph) for each patient.

**Table 1 healthcare-09-00407-t001:** Descriptive statistics for submandibular push exercise success and fail group.

Variables	Success	Fail	*p*-Value ^†^
Number (n)	8	4	-
Gender (n)			-
Male	7	4	
Female	1	0	
Age (years)	64.13 ± 14.47	57.00 ± 18.12	0.497
Disease duration (days)	39.00 ± 26.54	57.50 ± 33.56	0.443
Initial behavior parameter			
BBS	15.50 ± 13.75	0.00 ± 0.00	0.014 *
MMSE	21.75 ± 4.55	21.00 ± 2.70	0.347
MBI	31.13 ± 25.93	4.75 ± 5.25	0.124
BDI	26.57 ± 14.42	18.00 ± 24.04	0.558
PAS	4.17 ± 3.48	4.75 ± 3.30	0.655
VDS	33.00 ± 20.13	35.62 ± 19.33	0.915

Mean ± standard deviation; BBS, Berg balance scale; MMSE, mini-mental state examination; MBI, Modified Barthel Index; BDI, Beck Depression Inventory; PAS, Penetration–Aspiration Scale; VDS, Videofluoroscopic Dysphagia Scale; ^†^
*p*-value, comparison between success and fail group; * *p*-value < 0.05.

**Table 2 healthcare-09-00407-t002:** Comparison of the number of exercise, maximum pressure, area of pressure-time graph for each exercises.

Variables		Mean ± SD	*p*-Value ^a^	*p*-Value ^b^	*p*-Value ^c^	*p*-Value ^d^
Count (n)	D1	10.00 ± 2.20	0.861	0.332	0.461	0.546
	D2	10.12 ± 1.72	-	0.340	0.234	0.581
	D3	9.62 ± 1.768	-	-	0.023 *	0.167
	D4	10.63 ± 1.506	-	-	-	0.832
	D5	10.50 ± 3.071	-	-	-	-
Fmax (g)	D1	87.29 ± 31.16	0.028 *	0.012 *	0.017 *	0.012 *
	D2	106.01 ± 29.27	-	0.012 *	0.012 *	0.012 *
	D3	137.32 ± 31.53	-	-	0.889	0.050
	D4	143.78 ± 39.13	-	-	-	0.123
	D5	167.77 ± 37.39	-	-	-	-
Area (g·sec)	D1	1033.19 ± 609.56	0.025 *	0.025 *	0.025 *	0.012 *
	D2	1530.04 ± 573.82	-	0.123	0.093 *	0.012 *
	D3	1772.32 ± 762.61	-	-	0.674	0.036 *
	D4	1810.33 ± 791.13	-	-	-	0.050
	D5	2212.13 ± 548.62	-	-	-	-

SD—standard deviation; Count—the number of exercise; Fmax—maximum pressure; Area—the area of pressure-time graph; D1, 2, 3, 4, 5—dysphagia therapy (submandibular push exercise) trial times; ^a^
*p*-value, comparison with D2; ^b^
*p*-value, comparison with D3; ^c^
*p*-value, comparison with D4; ^d^
*p*-value, comparison with D5; * *p*-value < 0.05.

## Data Availability

Available upon reasonable request.

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
