# Peer review of "Submandibular Push Exercise Using Visual Feedback from a Pressure Sensor in Patients with Swallowing Difficulties: A Pilot Study"

_healthcare, 2021, doi:10.3390/healthcare9040407_

Round 1

Reviewer 1 Report

Overall I believe this is an interesting study, however it needs to be  stated how this study is novel or different than the previously published study using this exercise.

Introduction:

When it says "despite the usefulness of this feedback, it was only evaluated during one session" was the use of the visual feedback in only one session for the previously published work cited by reference 18, or does the current study only have visual feedback during one session? This just needs to be more clearly stated in this sentence.

Methods:

Please state how success vs failure was classified.

Please define the protocol more clearly. 5 sessions per week for 3 weeks means 15 sessions? How long did each session last? only 1 minute? Were there multiple trials in each session? How many? Says patient maintained for 3 seconds how long did the patient rest in between?

Table 1: please include list of abbreviations for all acronyms in the table legend.

Results: These are very confusing to me I think because I am confused about the protocol. Once you specify the protocol I believe I will understand the data. What does D mean in table 2 and figure 2? What does count mean? is this the number of trials? exercises? What does each dot represent in the bar graphs? each patient?

Discussion: You stated that post swallowing test were included in this study? how soon after completion of this study? Why was this data not included in the results?

Reviewer 2 Report

In this novel study, these authors describe the effects of a submandibular push exercise with visual biofeedback to improve swallow functioning in dysphagic neurologically impaired patients. The writing is clear, organized, and concise. Background and lit review is adequate. The Tabs and Figs are clear, other than my comments below. The Methodology section needs to be expanded, per my comments below. Results are clear, as is the discussion section. Limitations are adequate and appropriate. This is an interesting study that potentially adds to our toolbox of exercises to improve muscle function and swallowing. A few comments:

  1. How was dysphagia defined on VFSS? was a specific protocol used during VFSS? Were there multiple clinicians performing the VFSS? Was any inter/intra-judge reliability performed on the (assumedly) recorded VFSS studies? Please clarify.
  2. The authors need to describe the submandibular push exercises for the readers. In addition, they should add what specific instructions were given to the patients. It is not clear what exactly this exercise is – is it pushing down with the mandible? up with the mandible? is it the tongue doing the pushing? the authors need to describe this exercise in detail. This study cannot be replicated, nor can other clinicians utilize this exercise without this information.
  3. Where precisely is the pressure sensor placed? Please clarify.
  4. Several variables noted in the Results section were not described or included (other than the MMSE-K) in the Methodology section (i.e., BBS, MBI, PAS, VDS), nor were they referenced. Scales used in this study need to be added to the Methodology Section and need to be described briefly. What are the Videofluoroscopic Dysphagia Scale and Berg Balance Scales? Please define all scales used in this study.
  5. What were the diagnoses of the 4 patients who were unable to perform this exercise? The authors state that MBI was the factor that affected success with this exercise. Which specific factors related to the MBI were significant in these 4 patients? 
  6. Table 1 - In the footnotes, the authors should provide and spell out the words the abbreviations stand for in this table. The same goes for Figures 2-4 – please spell out “D”, “Fmax” in the footnotes.

Author Response

Thank you. 
